# Comparison of Modified Labor Induction Strategies for Pregnant Women at a Single Tertiary Center Before and During the COVID-19 Pandemic

**DOI:** 10.3390/diagnostics14232739

**Published:** 2024-12-05

**Authors:** Yi-Sin Tan, Ching-Chang Tsai, Hsin-Hsin Cheng, Yun-Ju Lai, Pei-Fang Lee, Te-Yao Hsu, Kun-Long Huang

**Affiliations:** 1Department of Obstetrics and Gynecology, Kinmen Hospital, Ministry of Health and Welfare, Jinhu 891002, Taiwan; tysin800201@gmail.com; 2Department of Obstetrics and Gynecology, Kaohsiung Chang Gung Memorial Hospital and Chang Gung University College of Medicine, Kaohsiung 833401, Taiwan; aniki@cloud.cgmh.org.tw (C.-C.T.); chokovarous@cloud.cgmh.org.tw (H.-H.C.); lusionbear@cloud.cgmh.org.tw (Y.-J.L.); pf7938@cgmh.org.tw (P.-F.L.); tyhsu@cgmh.org.tw (T.-Y.H.)

**Keywords:** COVID-19 pandemic, labor induction, cesarean section, maternal outcomes, fetal outcomes

## Abstract

Background: The COVID-19 pandemic has substantially impacted healthcare systems and obstetric practices worldwide. Labor induction is a common procedure for preventing obstetric complications in high-risk populations. This study evaluated perinatal outcomes of labor induction using a modified management protocol in a tertiary care center during the COVID-19 pandemic. Methods: We conducted a retrospective study by reviewing electronic structured delivery records of women who underwent elective labor induction between June 2020 and October 2022. We analyzed maternal characteristics, maternal outcomes, and neonatal outcomes during the pre-pandemic (June 2020 to May 2021) and pandemic periods (May 2021 to October 2022). Results: The study included 976 cases: 325 pregnancies in the pre-pandemic group and 651 in the pandemic group. The pandemic group showed earlier gestational age at delivery (39 vs. 40 weeks, *p* < 0.01) and lower body mass index (27.1 vs. 27.5 kg/m^2^, *p* = 0.03). During the pandemic period, we observed a significant increase in labor induction cases and a decrease in cesarean sections. Neonatal outcomes, including Apgar scores and intensive care admissions, showed no significant differences between groups. Subgroup analysis identified advanced maternal age (OR = 1.08; 95% CI = 1.03–1.14; *p* < 0.01) and primiparity (OR = 5.24; 95% CI = 2.75–9.99; *p* < 0.01) as independent risk factors for cesarean delivery. Conclusions: Even under modified protocols for labor induction during the COVID-19 pandemic, more pregnancies underwent labor induction while achieving a significant reduction in cesarean sections. Advanced maternal age and primiparity were identified as independent risk factors associated with cesarean delivery.

## 1. Introduction

Labor induction has been a common obstetric practice aimed at preventing complications during pregnancy, such as post-term pregnancies, fetal growth restriction, and maternal medical conditions. However, the COVID-19 pandemic raised concerns about the safety and effectiveness of induction protocols due to increased infection risks and related challenges. Studies have shown that various induction methods, including planned inductions for low-risk pregnancies and telemedicine for high-risk pregnancies, can reduce COVID-19 transmission risks and improve outcomes for both mothers and fetuses [1,2]. Nevertheless, concerns have been raised about potential increases in induction failure rates and the need for additional interventions to achieve successful labor outcomes [3].

In December 2019, a series of pneumonic infections were identified in China, with the pathogen confirmed as the 2019 novel coronavirus. The illness was officially named “Coronavirus Disease 2019” (COVID-19) on 11 February 2020 [4]. According to Taiwan Centers for Disease Control (CDC) data, the COVID-19 pandemic in Taiwan experienced two major waves. The first wave occurred from May to July 2021, involving both imported cases and local community transmission. A second surge, driven by the omicron variant, emerged in January 2022, with monthly case numbers peaking in May. As of 20 July 2023, Taiwan reported 10,241,506 confirmed COVID-19 cases, including 19,005 deaths [5].

Pregnant women represented a high-risk population during the COVID-19 pandemic due to potential adverse effects on both maternal and fetal health. Evidence indicates that pregnant women with COVID-19 face elevated risks of severe complications, including hospitalization, intensive care unit (ICU) admission necessitating mechanical ventilation or extracorporeal membrane oxygenation (ECMO), and mortality [6,7]. COVID-19 infection during pregnancy is associated with increased incidence of preterm birth, preeclampsia, and stillbirth compared to uninfected pregnancies. Furthermore, infants born to COVID-19-positive mothers show higher rates of neonatal intensive care unit (NICU) admission and respiratory distress syndrome [8]. These findings emphasize the critical importance of enhanced surveillance for pregnant women with COVID-19 to optimize maternal and neonatal outcomes.

The pandemic has prompted widespread modifications to perinatal care protocols across various countries, primarily driven by the need to minimize direct physical contact. Notable adaptations include the increasing adoption of outpatient cervical ripening protocols in the United Kingdom [2,9]. Telemedicine has emerged as a crucial innovation in antenatal care, enabling virtual consultations that reduce in-person contact and potential virus exposure. While this transition has generally proved beneficial, it presents challenges, particularly regarding the limitations in physical examination and monitoring: essential components of comprehensive antenatal care [3,10].

Studies examining the impact of increased induction rates during the pandemic have reported variable outcomes: some demonstrate higher cesarean section rates [3], others show no significant change [11,12,13], and some indicate reduced cesarean section rates following labor induction [14]. Given these inconsistent findings, our study aims to evaluate maternal and neonatal outcomes under a modified labor induction protocol implemented during the COVID-19 pandemic at a single tertiary center.

## 2. Materials and Methods

Prior to the COVID-19 pandemic, at Kaohsiung Chang Gung Memorial Hospital (KCGMH), labor induction was routinely offered to low-risk pregnancies at or beyond 39 weeks of gestation and to high-risk pregnancies at or beyond 38 weeks of gestation. Obstetric conditions that increased fetal and maternal risks included intrauterine growth restriction, prelabor rupture of membranes, hypertensive disorders, diabetes, chorioamnionitis, and fetal demise. In May 2021, we implemented a comprehensive admission and scheduled induction protocol aligned with epidemic prevention guidelines from the Taiwan CDC [15]. The protocol for scheduled labor induction is illustrated in Figure 1. Compared to our standard care for labor induction before COVID-19 pandemic, pregnant women underwent COVID-19 real-time polymerase chain reaction (RT-PCR) testing at the outpatient clinic before admission for planned induction. With a negative test result, labor induction could be scheduled within 48–72 h. Prior to induction of labor, all patients were required to have a negative COVID-19 RT-PCR test and undergo evaluation of fetal biophysical profiles (including fetal movement, fetal breathing movement, fetal tone, amniotic fluid volume, and a nonstress test), pelvic examination, and blood tests. Labor induction was scheduled when patients met the following criteria: fetal biophysical profile score ≥ 8, unfavorable cervical effacement, and normal results for complete blood count, liver function tests, renal function tests, and coagulation profile. Pregnant women with positive COVID-19 RT-PCR results underwent home isolation unless they experienced obstetric emergencies. Those requiring urgent medical attention were evaluated in our emergency department and transferred to the isolation zone for further management. For those with active obstetric concern, chest X-rays were performed to assess the severity of pulmonary involvement, and pulmonologists were consulted for coordinated care.

Upon admission, each woman underwent vaginal examination to assess cervical status, including cervical dilation, effacement, and fetal station. Routine fetal heart monitoring was maintained throughout labor induction. For women with an unfavorable cervix, Dinoprostone (Prostin^®^ E2 VT 3 mg) vaginal inserts were administered every six hours, not exceeding the maximum daily dose of 6 mg. Oral misoprostol (Cytotec^®^ 200 mcg) was another option, administered at a dose of 25 mcg every two hours in accordance with the World Health Organization recommendations (2022) [16]. Both drugs were never administered concurrently. Oxytocin augmentation was administered when cervical dilation exceeded three centimeters. Artificial rupture of membranes was not performed during labor induction. Failed induction was defined as the inability to establish regular uterine contractions, failure to enter active labor (exceeding 48 h), fetal distress, or maternal morbidity.

### 2.1. Data Collection

We conducted a retrospective review of our hospital’s electronic structured delivery database, which contains admission reasons, prenatal examination findings, and patient characteristics. The institutional review board approved this study (number: 202300953B0). Patient identifiers such as names and addresses were excluded from the data extraction process, thereby ensuring patient anonymity and protecting maternal privacy.

The study included pregnant women who underwent elective labor induction at KCGMH between 1 June 2020, and 1 October 2022. KCGMH serves as a tertiary medical center in southern Taiwan. We defined the pre-pandemic period as 1 June 2020 through 14 May 2021, and the pandemic period as 15 May 2021 to 1 October 2022, based on Taiwan’s transition to community-wide COVID-19 transmission. After 1 October 2022, Taiwan CDC eliminated mandatory COVID-19 RT-PCR testing for hospital admissions.

Exclusion criteria comprised previous cesarean deliveries, multiple pregnancies, deliveries before 38 weeks’ gestation, inductions prompted by maternal medical conditions or fetal anomalies, and cases with incomplete data. We analyzed baseline patient characteristics, delivery information, maternal outcomes (delivery mode, immediate postpartum hemorrhage, severe perineal laceration, maternal death), and neonatal outcomes (5-min Apgar score, neonatal intensive care unit admission, and birth weight exceeding 4000 g).

### 2.2. Statistical Analysis

Data analysis was performed using IBM SPSS software version 22 (SPSS Inc., Chicago, IL, USA). We evaluated data distribution normality using the Kolmogorov–Smirnov test and compared continuous variables between groups using the Mann–Whitney U-test. For categorical data analysis, we employed the chi-square test for samples larger than 20 and Fisher’s exact test for samples of 20 or fewer. We compared parity and obstetric complications (including pregnancy-induced hypertension/preeclampsia and gestational diabetes mellitus), maternal outcomes (delivery mode, postpartum hemorrhage, 3rd/4th-degree laceration), and fetal outcomes (neonatal intensive care unit admission and macrosomia) using the chi-square test. Maternal medical conditions (hypertension, diabetes mellitus, and autoimmune diseases) and fetal outcomes (5-min Apgar < 7) were compared using Fisher’s exact test. We presented gestational age at delivery, maternal age, maternal height, maternal weight, and body mass index (BMI) as median (minimum to maximum). To adjust for clinical characteristic imbalances between vaginal delivery and cesarean section groups, we conducted multivariate analysis using a logistic regression model. Statistical significance was defined as *p* < 0.05.

## 3. Results

### 3.1. Patient Characteristics

Of 3513 deliveries between 1 June 2020 and 1 October 2022, 976 women met the inclusion criteria for labor induction admission. We excluded 37 cases: 26 for induction before 38 weeks of gestation, six for multiple pregnancies, one for maternal medical problems, one for fetal anomalies, and three for incomplete data. The final analysis included 325 births in the pre-pandemic period and 651 births in the pandemic period (Figure 2).

Table 1 summarizes baseline patient characteristics. Significant differences between groups were observed in gestational age at delivery (*p* < 0.01) and BMI (*p* = 0.03). The pandemic group showed earlier delivery timing compared to the pre-pandemic group (39 weeks [60%] vs. 40 weeks [46%], *p* < 0.01). Pre-pandemic participants had higher maternal BMI compared to pandemic participants (27.5 vs. 27.1, *p* = 0.03). Over 60% of deliveries in both groups were primigravida. No significant differences were found in age, height, weight, maternal medical conditions, and obstetric complications.

### 3.2. Maternal and Fetal Outcomes

During the pandemic period, we observed a significant increase in monthly labor induction cases (39 vs. 28 cases/month, *p* < 0.05) and a decrease in cesarean sections following labor trial (13.21% vs. 18.77%, *p* = 0.02) (Table 2). Postpartum hemorrhage incidence and third/fourth-degree perineal laceration rates showed no significant differences. Neonatal outcomes, including 5-min APGAR scores < 7, NICU admission rates, and large birth weight (≥4000 g) occurrence, remained comparable between groups.

### 3.3. Subgroup Analyses

Table 3 presents factors associated with maternal and neonatal outcomes during the pandemic period. Univariate analysis showed no significant associations between delivery mode and height, weight, pregnancy-induced hypertension (PIH), gestational diabetes mellitus (GDM), maternal medical conditions, or macrosomia. However, later gestational age at delivery (39.6 vs. 39.5 weeks, *p* < 0.01), advanced maternal age (34 vs. 33 years, *p* = 0.02), higher BMI (28.5 vs. 27.5, *p* = 0.02), and primigravida status (86% vs. 58%, *p* < 0.01) were significantly associated with cesarean section after labor trial. Multivariate analysis identified advanced maternal age (OR = 1.08; 95% CI = 1.03–1.14; *p* < 0.01) and primigravida status (OR = 5.24; 95% CI = 2.75–9.99; *p* < 0.01) as independent factors associated with increased cesarean section rates.

## 4. Discussion

Our study revealed that the COVID-19 pandemic led to increased preference for scheduled inductions over spontaneous labor. We observed a significantly reduced cesarean delivery rate following labor trial in the pandemic group compared to the pre-pandemic group, consistent with the findings of Sinnott et al. [1]. The pandemic group showed an increase in labor inductions at 39 weeks without corresponding increases in adverse perinatal outcomes, potentially validating real-world application of the ARRIVE trial results [17]. The previous literature examining COVID-19’s impact on obstetric interventions has produced varied results. Multiple international studies have assessed the pandemic’s influence on intervention rates, including induction, cesarean births, and operative vaginal births, with impacts varying widely across countries. Several studies reported no significant changes in these rates [11,12,13]. However, Canadian data showed increased obstetric interventions, including labor induction and cesarean section, reflecting heightened stress and uncertainty during the pandemic [3]. Conversely, an Icelandic study demonstrated decreased overall cesarean rates [14]. A retrospective study across 15 U.S. hospitals found no statistically significant changes in intrapartum interventions or delivery modes following pandemic onset [18]. Sinnott et al. reported decreased cesarean births alongside increased induction rates [1], aligning with our findings.

The American College of Obstetricians and Gynecologists (ACOG) and the Society for Maternal-Fetal Medicine (SMFM) have advised against elective, nonmedically indicated deliveries before 39 weeks gestation since 2009, aiming to minimize adverse neonatal outcomes, including NICU admissions and respiratory complications [19]. However, we observed increased labor inductions at 38 weeks during the pandemic period. A review of medical records revealed primary induction reasons as maternal anxiety, pregnancy discomfort, macrosomia concerns, and cesarean section risk [20]. While guidelines discourage pre-39-week induction, our pandemic-period observations showed no increase in adverse neonatal outcomes for 38-week inductions.

Jacob et al. demonstrated that higher BMI significantly increased cesarean section risk during labor induction [21]. A population-based cohort study also showed obesity’s association with increased labor induction and failure rates [22]. Another retrospective cohort study conducted on more than 40,000 pregnant women to stratify BMI classes demonstrated that when BMI was in the overweight category (>25 kg/m^2^), the cesarean section rate was 1.5 times higher than in women with normal BMI [23]. Among nulliparous pregnancies with BMI > 40 kg/m^2^, more than 50% resulted in cesarean sections following labor induction [24]. The higher induction failure rate among obese pregnancies can be attributed to abnormal metabolism in uterine smooth muscle, as demonstrated in obese mice models showing increased long-chain fatty acids, resulting in dysfunctional myometrial contractility [25].

In recent decades, the trend of women prioritizing professional achievements and lifestyle choices over early marriage has contributed to an increase in advanced maternal age pregnancies. Advanced maternal age has been proven to increase the risks of stillbirth, maternal hypertension, gestational diabetes, preterm birth, placental abnormalities, and poor neonatal outcomes [26,27]. Research has shown increased cesarean rates with advanced maternal age, particularly for women over 40, regardless of parity [28]. A matched retrospective cohort study demonstrated threefold-increased cesarean risk with advanced maternal age (≥35 years) [29]. A similar retrospective cross-sectional study reported a 4.2-fold-higher risk of induction failure among women aged more than 35 years [30]. In a prospective study analyzing specific pregnancy populations, advanced maternal age (>34 years) emerged as a significant independent risk factor for failed induction requiring cesarean delivery [31]. In another Australian study, where advanced maternal age was defined as greater than 38 years, researchers reported that these women had a two-fold higher likelihood of cesarean delivery [32]. While this discussion may not be directly related to the study objectives, we believe that advanced maternal age significantly impacts labor induction outcomes, independent of infectious disease status.

According to World Health Organization recommendations (2022), labor induction in term pregnancies can be initiated using any of the following methods to stimulate uterine contractions: low-dose vaginal misoprostol (25 mcg every 6 h), oral misoprostol (25 mcg every 2 h), low-dose vaginal prostaglandins, intravenous oxytocin, or cervical balloon dilators [16]. Normal labor progression is characterized by cervical ripening and dilation under regular uterine contractions with reassuring fetal heart rate patterns within an expected timeframe [33]. For nulliparous women, the possible mechanisms explaining higher induction failure rates include less sensitivity to ripening agents and an undilated pelvis. Previously published studies have demonstrated that primigravid status is an independent risk factor for failed induction [34,35,36]. These findings align with our results.

Pregnancies complicated by COVID-19 infection carry higher risks of adverse maternal and neonatal outcomes, including respiratory failure, intensive care unit admission, pregnancy loss, stillbirth, intrauterine growth restriction, preterm birth, and changes in delivery protocols [37]. Cytokine storms may induce severe pregnancy complications [38]. Pregnant women in their third trimester are more susceptible to COVID-19 infection compared to those in their first trimester. A retrospective study reported approximately 34% admission rates for pregnancies during the third trimester compared to 7% during the first trimester [39]. To prevent perinatal transmission and maternal cardiopulmonary dysfunction, cesarean delivery is often preferred by clinicians [40]. In our study, while the included cases were admitted for labor induction without COVID-19 infection, records of COVID-19 infection after admission were not available. Therefore, we were unable to analyze the relative risks of induction failure and adverse maternal and neonatal outcomes among this group.

During the COVID-19 pandemic, adverse neonatal outcomes were reported, including fever, respiratory distress, NICU admission, preterm birth, and white blood cell disturbances [41,42,43]. While obstetric interventions decreased, there was a transient increase in perinatal mortality during the pandemic period [44]. Racine et al. conducted a retrospective study in a single center with low COVID-19 incidence to analyze how policy changes impacted perinatal outcomes. The ICU admission and readmission rates for both mothers and neonates did not reach statistical significance. They observed increased spontaneous labor and shorter hospital stays [45]. A multicenter retrospective cohort study showed no significant trend toward increased risk of macrosomia [46]. These findings suggest that factors associated with adverse neonatal outcomes were primarily influenced by maternal-fetal conditions and may have been further impacted by hospital policies during the COVID-19 pandemic.

Our modified protocol focused on scheduling labor induction 48–72 h after COVID-19 RT-PCR testing to prevent nosocomial COVID-19 infection. The current literature review suggests that outpatient management using cervical mechanical dilators can minimize hospital stay and reduce medical costs. Mechanical ripening agents for low-risk pregnancies resulted in significantly shorter hospital stays [47]. Saunders et al. developed a cost–consequence model for outpatient management using hygroscopic cervical dilators, which showed results similar to the above study, indicating reduced nursing time [48]. While outpatient oral misoprostol for cervical ripening was also offered to reduce admission time, higher rates of chorioamnionitis and unplanned home births were observed [49,50]. A systematic review of outpatient induction using slow-release dinoprostone vaginal inserts demonstrated improved labor induction experience but required more oxytocin augmentation compared to the inpatient group [51]. For low-risk pregnancies where patients prefer shorter hospital stays and reduced COVID-19 exposure risk, these outpatient management strategies could be incorporated into our protocol.

Our study’s strengths include the established pandemic-period induction protocol ensuring consistent practice and comprehensive structured delivery database documentation. Based on our findings, in the event of future COVID-19 surges or other infectious disease pandemics, pregnant women of advanced maternal age and nulliparous patients should be counseled about their increased risk of failed induction. However, limitations include lack of induction medication records, absence of COVID-19 infection severity impact assessment, retrospective nature, and single-center scope limiting result generalizability. Further multicenter prospective studies are warranted.

## 5. Conclusions

Cesarean delivery rates following labor induction significantly decreased during the COVID-19 pandemic, while maternal complications and adverse neonatal outcomes remained stable under the modified protocol. Advanced maternal age and nulliparity emerged as independent factors which increased cesarean section rates in the pandemic group.

## Figures and Tables

**Figure 1 diagnostics-14-02739-f001:**
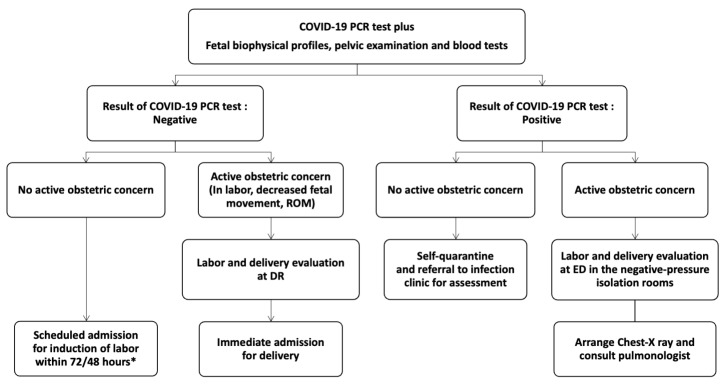
Modified protocol for scheduled induction of labor. Labor induction was scheduled when patients met the following criteria: fetal biophysical profile score ≥ 8, unfavorable cervical effacement, and normal results for complete blood count, liver function tests, renal function tests, and coagulation profile. PCR, Polymerase Chain Reaction; ROM, rupture of membrane; DR, delivery room; ED, emergency department. * For appointments before 3 May 2022, the timeframe is set at 72 h, while for appointments after 4 May 2022, it is set at 48 h. These policies were based on guidelines from the Taiwan Centers for Disease Control.

**Figure 2 diagnostics-14-02739-f002:**
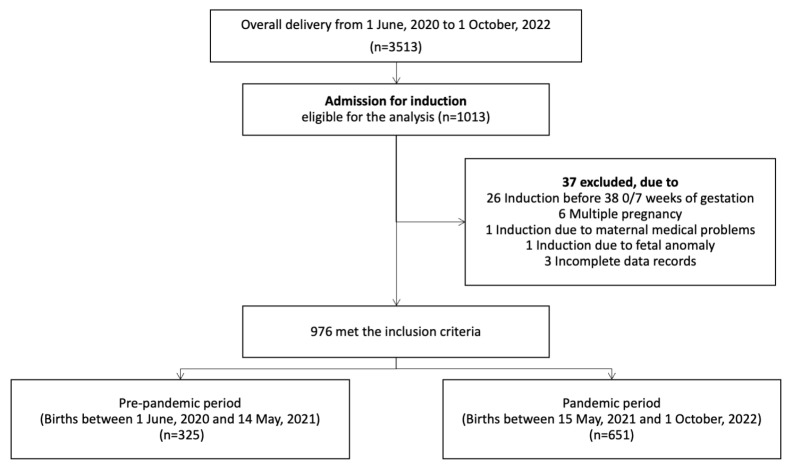
Flow diagram for patient selection.

**Table 1 diagnostics-14-02739-t001:** Characteristics of the study population.

Characteristic	Pre-Pandemic Period		Pandemic Period		*p* Value
	Median (Range)	No. (%)	Median (Range)	No. (%)	
Overall	-	325	-	651	
Gestational age at delivery (wk)	40.0 (38.0–41.6)	-	39.4 (38.0–41.9)	-	<0.01 *
38	-	30 (9)	-	109 (17)	
39	-	126 (39)	-	391 (60)	
40	-	151 (46)	-	135 (21)	
41	-	18 (6)	-	16 (2)	
Maternal age (y)	33 (17–44)	-	33 (19–50)	-	0.25
34 or younger	-	200 (62)	-	385 (59)	
35 or older	-	125 (38)	-	266 (41)	
Height (cm)	161 (145–181.6)	-	161 (144.4–182)	-	0.43
Weight (kg)	71 (48.1–130.8)	-	70 (50–113.5)	-	0.08
Body mass index (kg/m^2^)	27.5 (18.3–46.3)	-	27.1 (19.4–36)	-	0.03 *
<30	-	239 (73)	-	482 (74)	
30–39	-	81 (25)	-	165 (25)	
≥40	-	5 (2)	-	4 (1)	
Parity					0.84
Nulliparous	-	199 (61)	-	403 (62)	
Multiparous	-	126 (39)	-	248 (38)	
Medical condition					
Hypertension	-	5 (1.5)	-	7 (1.1)	0.55
Diabetes mellitus	-	4 (1.2)	-	6 (0.9)	0.74
Autoimmune disease	-	7 (2.2)	-	8 (1.2)	0.28
Obstetric complication					
PIH/Pre-eclampsia	-	7 (2.2)	-	18 (2.8)	0.57
GDM	-	27 (8.3)	-	72 (11.1)	0.18

PIH, pregnancy-induced hypertension; GDM, gestational diabetes mellitus. * *p* < 0.05.

**Table 2 diagnostics-14-02739-t002:** Maternal outcomes and perinatal outcomes between study groups.

	Pre-Pandemic Period		Pandemic Period		*p* Value
	No.	%	No.	%	
Maternal outcomes					
Induction numbers per month. Delivery mode	28		39		0.01 *
Vaginal delivery	264	81.23%	565	86.79%	
Cesarean section	61	18.77%	86	13.21%	0.02 *
Postpartum hemorrhage	6	1.85%	16	2.46%	0.54
3rd/4th-degree laceration	18	5.54%	39	5.99%	0.77
Death	0	0.00%	0	0.00%	NA
Fetal outcomes					
5-min Apgar < 7	2	0.62%	3	0.46%	1
Neonatal intensive care unit admission	20	6.15%	26	3.99%	0.13
Macrosomia (4000 g or more)	8	2.46%	14	2.15%	0.76

NA, not applicable. * *p* < 0.05.

**Table 3 diagnostics-14-02739-t003:** Factors related to the outcomes of induction of labor during the COVID-19 pandemic.

Characteristic	Vaginal Delivery		Cesarean Section		*p* Value
	Median (Range)	No. (%)	Median (Range)	No. (%)	
Overall	-	565	-	86	
Gestational age at delivery (wk)	39.5 (38.0–41.9)	-	39.6 (38.1–41.0)	-	<0.01 *
38	-	99 (17)	-	10 (12)	
39	-	344 (61)	-	47 (55)	
40	-	107 (19)	-	28 (32)	
41	-	15 (3)	-	1 (1)	
Maternal age (y)	33 (20–50)	-	34 (19–46)	-	0.02 *
34 or younger	-	343 (61)	-	42 (49)	
35 or older	-	222 (39)	-	44 (51)	
Height (cm)	161 (144.4–182)	-	160.9 (145–175)	-	0.28
Weight (kg)	69.9 (50–113)	-	71 (52–113.5)	-	0.14
Body mass index (kg/m^2^)	27.5 (19.4–46.1)	-	28.5 (20.3–39.8)	-	0.02 *
<30	-	419 (74)	-	62 (72)	
30–39	-	141 (25)	-	24 (28)	
≥40	-	5 (1)	-	0 (0)	
Parity					<0.01 *
Nulliparous	-	329 (58)	-	74 (86)	
Multiparous	-	236 (41)	-	12 (14)	
Medical condition					
Hypertension	-	7 (1.2)	-	0 (0)	0.6
Diabetes mellitus	-	4 (0.7)	-	2 (2.3)	0.18
Autoimmune disease	-	6 (1.1)	-	2 (2.3)	0.29
Obstetric complication					
PIH/Pre-eclampsia	-	15 (2.7)	-	3 (3.5)	0.72
GDM	-	58 (10.3)	-	14 (16.3)	0.1
Fetal outcome					
5-min Apgar <7	-	2 (0.4)	-	1 (1.2)	0.35
Neonatal intensive care unit admission	-	21 (3.7)	-	5 (5.8)	0.36
Macrosomia (4000 g or more)	-	10 (1.8)	-	4 (4.7)	0.1

PIH, pregnancy-induced hypertension; GDM, gestational diabetes mellitus. * *p* < 0.05.

## Data Availability

There are no additional data on any other site other than in this manuscript.

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
