# Peer review of "Comparison of Modified Labor Induction Strategies for Pregnant Women at a Single Tertiary Center Before and During the COVID-19 Pandemic"

_diagnostics, 2024, doi:10.3390/diagnostics14232739_

Round 1
Reviewer 1 Report
Comments and Suggestions for Authors
In the study ''Comparison of modified labor induction strategies for pregnant women at a single tertiary center before and during the 3 COVID-19 pandemic'', Tan YS et al., evaluated perinatal outcomes of labor induction a retrospective study realized between June 2020 and October 2022.
Altough this article is well structured and write , I consider that the topic isn't so actual at this time. But this manuscript is valuable due to its potential to be helpful in reviewing national and international guidelines and recommendations regarding conduit of protocols in special pandemic conditions.
In the final of the article, the strengths and limitations of the study are mentioned. I believe that a certain limitation of the analyzed data is represented by the lack of more detailed reports on the diagnostic methods (blood tests or other specific parameters evaluated upon admission that indicated the decision to induce labor). The manuscript can be improved in order to fit with the topic of the journal Diagnostic, by adding some details on the protocol used in the investigations performed upon admission to the hospital (not only the PCR test that is mentioned). What types of tests were performed? blood tests? ultrasound? what parameters were modified for each batch analyzed? Were there modified biochemical parameters that could be used as indicators for following the described protocol?
After minor revision, I strongly suggest publication of this article.
Author Response
Comments 1: In the final of the article, the strengths and limitations of the study are mentioned. I believe that a certain limitation of the analyzed data is represented by the lack of more detailed reports on the diagnostic methods (blood tests or other specific parameters evaluated upon admission that indicated the decision to induce labor). The manuscript can be improved in order to fit with the topic of the journal Diagnostic, by adding some details on the protocol used in the investigations performed upon admission to the hospital (not only the PCR test that is mentioned). What types of tests were performed? blood tests? ultrasound? what parameters were modified for each batch analyzed? Were there modified biochemical parameters that could be used as indicators for following the described protocol?
Response 1: We have added more detailed reports including fetal biophysical profiles, blood test results, and pelvic examination findings in line 91-97. These assessment parameters remained unchanged before and during the COVID-19 pandemic. The only addition was the requirement of a COVID-19 PCR test prior to admission. For patients with active COVID-19 infection and signs of labor, we performed chest X-rays to assess lung involvement/severity and consulted pulmonologists for coordinated care (revised in line 102-104). The figure 1 was also revised based on above changes.
Comments 2: After minor revision, I strongly suggest publication of this article.
Response 2: Thank you very much for your positive feedback and recommendation.
Reviewer 2 Report
Comments and Suggestions for Authors
This is a retrospective review of induction of labour comparing two groups of mothers admitted to a tertiary maternity centre before and during the Covid-19 pandemic. Data is derived from electronic records and ethical approval has been obtained. The authors should elaborate about data extraction methods and anonymization methods employed for this study.
The abstract, Introduction and objectives of the study are written with clarity.
Under Methods, the authors should state clearly what modification was done to induction of labour protocols and quote the standard of care offered to mothers in the institute. Line 103 relates to use of Dinoprostone and Misoprostol without providing details about how these medications were prescribed. Kindly detail the standard method employed, if both drugs were used simultaneously and when augmentation ( and ARM) were done. Were patient's informed ( consent) about the risks and complications of the method of induction? Mechanical methods ( hygroscopic ) is stated in the Discussion. Foley catheter induction of labour is also an option. Were these methods of induction available to done in the institute?
What is meant by failed method of induction? Kindly provide some details and if time-duration and maternal anxiety were factors that led to cesarean deliveries (CD).
Line 118: Under exclusion criteria, kindly state if patients with previous CD were excluded.
The tables showing the results of the variables study are well laid out and the information is easy to read and understand.
Discussion:
This study does show some differences between pre-Covid and during the pandemic. Having said that, how does these results impact the current management of cases. Some critical comments about the relevance of this study would be useful.
The discussion about older mothers may not be relevant to the objectives of the study. Similarly, the generic approach to a discussion of induction methods seems out of place. It would be useful to quote current standard methods applied for effective induction of labour based on evidence ( Lines 232-238).
The limitations of the study should be stated; in view of the retrospective nature of the review.
References are well written.
Author Response
Comments 1: This is a retrospective review of induction of labour comparing two groups of mothers admitted to a tertiary maternity centre before and during the Covid-19 pandemic. Data is derived from electronic records and ethical approval has been obtained. The authors should elaborate about data extraction methods and anonymization methods employed for this study.
Response 1: Prior to initiating the study, our research protocol was submitted to the Institutional Review Board (IRB) of our hospital for review. After obtaining IRB approval, data were extracted from the structured electronic medical record system. Patient identifiers such as names and addresses were excluded from the data extraction process, thereby ensuring patient anonymity and protecting maternal privacy. (revised in line 133-135)
Comments 2: The abstract, Introduction and objectives of the study are written with clarity.
Response 2: Thank you very much for your positive feedback and recommendation.
Comments 3: Under Methods, the authors should state clearly what modification was done to induction of labour protocols and quote the standard of care offered to mothers in the institute.
Response 3: We had added more detailed standard care of labor induction in our institute in line 80-85. Modified protocol was emphasized clearly in line 87-88.
Comments 4: Line 103 relates to use of Dinoprostone and Misoprostol without providing details about how these medications were prescribed. Kindly detail the standard method employed, if both drugs were used simultaneously and when augmentation (and ARM) were done.
Response 4: We have added detailed information about the clinical administration of dinoprostone and misoprostol in lines 119-124. These medications were never administered concurrently. Artificial rupture of membranes was not performed during labor induction. (line 125-126)
Comments 5: Were patient's informed (consent) about the risks and complications of the method of induction?
Response 5: Yes, when admission for labor induction, patients were provided with a consent form from our institute detailing potential risks and complications. All patients were thoroughly counseled about possible risks and complications associated with labor induction before signing the informed consent. (line 97-99)
Comments 6: Mechanical methods (hygroscopic ) is stated in the Discussion. Foley catheter induction of labour is also an option. Were these methods of induction available to done in the institute?
Response 6: In our institute, mechanical methods for labor induction were not routinely used for term pregnancies and were reserved only for cases of fetal demise requiring termination.
Comments 7: What is meant by failed method of induction? Kindly provide some details and if time-duration and maternal anxiety were factors that led to cesarean deliveries (CD).
Response 7: We have already defined 'failed induction' in lines 126-128, including the time criterion (exceeding 48 hours) in line 127. Regarding maternal anxiety, this factor was not recorded in our structured electronic medical record system, preventing its inclusion in our analysis. We appreciate your suggestion and will consider incorporating maternal anxiety assessment in future studies examining factors associated with cesarean delivery.
Comments 8: Line 118: Under exclusion criteria, kindly state if patients with previous CD were excluded.
Response 8: We appreciate your suggestion and have added 'previous cesarean delivery' to our exclusion criteria in line 142.
Comments 9: The tables showing the results of the variables study are well laid out and the information is easy to read and understand.
Response 9: Thank you very much for your positive feedback.
Comments 10: Discussion: This study does show some differences between pre-Covid and during the pandemic. Having said that, how does these results impact the current management of cases. Some critical comments about the relevance of this study would be useful.
Response 10: Based on our findings, in the event of future COVID-19 surges or other infectious disease pandemics, pregnant women of advanced maternal age and nulliparous patients should be counseled about their increased risk of failed induction. This statement was added in line 310-312.
Comments 11: The discussion about older mothers may not be relevant to the objectives of the study. Similarly, the generic approach to a discussion of induction methods seems out of place. It would be useful to quote current standard methods applied for effective induction of labour based on evidence (Lines 232-238).
Response 11: Thank you for your suggestions. While this discussion may not be directly related to the study objectives, we believe that advanced maternal age significantly impacts labor induction outcomes, independent of infectious disease status (line 256-258). We also have added the WHO recommendations regarding induction methods in lines 257-261.
Comments 12: The limitations of the study should be stated; in view of the retrospective nature of the review.
Response 12: The limitation regarding the retrospective nature of this study has been addressed in line 314.
Comments 13: References are well written.
Response 13: Thank you very much for your positive feedback.